# Valorizing Astringent ‘Rojo Brillante’ Persimmon Through the Development of Persimmon-Based Bars

**DOI:** 10.3390/foods13233748

**Published:** 2024-11-22

**Authors:** Sepideh Hosseininejad, Gemma Moraga, Isabel Hernando

**Affiliations:** Instituto Universitario de Ingeniería de Alimentos—Food UPV, Universitat Politècnica de València, Camino de Vera, s/n, 46022 Valencia, Spain; sehos@doctor.upv.es (S.H.); gemmoba1@tal.upv.es (G.M.)

**Keywords:** *Diospyros kaki*, energy bars, astringency, tannins, carotenoids, antioxidant activity, in vitro digestion, sustainability

## Abstract

This study developed a new energy bar using the astringent ‘Rojo Brillante’ variety of persimmons to address postharvest losses. The bar was formulated with dehydrated persimmons, walnuts, hazelnuts, and chia seeds to enhance their nutritional profile. The proximate composition was evaluated and the mechanical and optical properties, soluble tannins, carotenoids, and antioxidant activities were monitored during storage. In addition, in vitro gastrointestinal digestion was performed to determine the recovery index of the bioactive compounds. The results showed that the formulated energy bar contained higher levels of healthy fats, proteins, and fibers than other fruit energy bars. The mechanical properties of dehydrated persimmon effectively supported the consistency of the bar, eliminating the need for hydrocolloids or syrups. During storage, soluble tannin content decreased, mitigating astringency issues commonly found in persimmon products, whereas carotenoid levels and antioxidant activity remained stable. In vitro digestion analysis revealed a higher recovery index for soluble tannins (180.08%) than carotenoids (9.87%). This persimmon-based energy bar offers a sustainable and nutritious option for the snack industry, catering to consumer preferences for natural products while contributing to the reduction of agricultural waste.

## 1. Introduction

Interest in enhancing the nutritional properties of many food products is increasing because of evolving eating patterns and increasing customer demand for nutritious foods with functional ingredients [1]. In the contemporary era, a considerable proportion of consumers prioritize the purchase of food items that align with their personal health and wellness goals. Consequently, the development of novel food products with distinctive textural, sensory, or functional attributes represents a significant area of interest [1].

Fruit bars are highly nutritious and calorically concentrated food products. Furthermore, they typically have a longer shelf life than raw fruits and can be a healthy, convenient food choice that provides dietary fiber and other bioactive substances needed to meet a person’s daily nutritional requirements [2].

Persimmon (*Diospyros kaki* Thunb.) is an important fruit crop in Spain, primarily cultivated on approximately 17,600 hectares, with the focus largely on the ‘Rojo Brillante’ variety, which represents more than 90% of production [3]. Persimmons are a rich source of bioactive compounds, including dietary fiber, minerals, vitamins, phenolic compounds, and carotenoids [4]. These compounds have been linked to several health benefits, including the prevention of certain diseases such as cancer, hypertension, diabetes, and atherosclerosis [5,6,7]. However, owing to their seasonal availability, perishable nature, and difficulties in storage and transportation, persimmons have experienced considerable postharvest losses in recent years [8]. In the case of astringent varieties, the losses were even more significant. The persimmon cultivar ‘Rojo Brillante’ is characterized by astringency due to its high soluble tannin content when harvested in a firm state [9]. As the fruit matures, the soluble tannin content declines [10]; however, the soft texture of the fruit renders it susceptible to rapid deterioration. Accordingly, postharvest deastringency treatment is required before commercialization, as fruits must be consumed when they still have a firm texture [10]. A variety of techniques have been used for deastringency treatment, including anaerobic treatments with CO2. This process precipitates tannins, rendering them undetectable when persimmon is consumed while simultaneously preserving the firmness of the product [11]. Nevertheless, the application of deastringency treatments represents an additional financial burden and may sometimes prove ineffective, engendering further losses.

To mitigate losses, persimmons can be processed (e.g., dried) to prolong their shelf life and serve as a promising functional ingredient for incorporation into novel food formulations. González et al. [12] demonstrated that hot-air drying has the effect of considerably reducing the amount of soluble tannins. The research team observed a significant reduction in soluble tannin content in dried persimmon slices after drying at 40 and 60 °C. These tannins undergo transformation into insoluble forms. It is thus possible to obtain non-astringent persimmon snacks from astringent fruits with no prior deastringency treatment. In addition, our previous study on the use of astringent persimmon flour as a functional ingredient in gluten-free muffins demonstrated the removal of astringency because of the insolubilization of tannins during baking [8]. Nevertheless, astringency issues have been documented in the context of various persimmon transformation processes. Castelló et al. [13] investigated the impact of thermal processing and storage on the astringency of a spreadable product derived from the ‘Rojo Brillante’ persimmon through osmotic processes. Despite undergoing a deastringency process and initial osmotic dehydration, which reduced the soluble tannin content, partial re-solubilization of insoluble tannins was observed when the product was subjected to elevated temperatures. Furthermore, during the storage period, an increase in the soluble tannin content was observed. It was determined that refrigeration was the optimal method to prevent astringency.

A review of the literature revealed no studies examining the production of persimmon-based bars. Several studies have been conducted on bars prepared from other dried fruits, including dates [14,15]. Ibrahim et al. [15] investigated the formulation of date-based bars with varying percentages of dates, ranging from 40 to 70%. The textural, sensory, and technological attributes of the bars were enhanced when 50% date paste was incorporated into the formulation, as reported by other researchers. In another study, Leguizamon-Delgado et al. [16] evaluated the physicochemical and sensory characteristics of mango-based bars prepared with dried mango as a viable alternative to agro-industrial waste. The results demonstrate the potential of this approach to use agro-industrial waste, with notable findings in terms of the total fiber content and functional compounds, including polyphenols and ascorbic acid, with antioxidant activity. Kumar et al. [17] used dried blends of papaya and guava for preparing fruit bars and reported that the optimal combination ratio was 50:50, as this combination yielded the highest sensory score, overall acceptability, and superior carotenoid and protein content.

Incorporating ingredients such as walnuts, hazelnuts, and chia seeds can help enhance the nutritional profile of fruit bars. These ingredients provide plant-based proteins and are rich in monounsaturated and polyunsaturated fats. Furthermore, they contain significant micronutrients, including minerals, vitamins, and other bioactive compounds, which augment the functional attributes of the final product. Walnuts are an exceptional source of omega-3 fatty acid content, particularly alpha-linolenic acid, and their consumption has been linked to health benefits and chronic disease prevention [18]. Hazelnuts are a notable source of vitamin E, a fat-soluble antioxidant, and are regarded as valuable ingredients in the food industry because of their distinctive flavor profile [19]. Chia seeds are a rich source of dietary fiber, particularly soluble fiber, which has been demonstrated to exert beneficial effects on the microbiota [20,21,22]. It is also noteworthy that the addition of dehydrated persimmons contributes to the cohesiveness of the fruit bar, resulting in a mix of textures and flavors when combined with nuts and seeds. Furthermore, dehydrated persimmon provides a natural sweetness that obviates the need for additional sweeteners.

The objective of this study was to obtain and characterize a new fruit bar formulated with persimmon of the astringent ‘Rojo Brillante’ variety as a strategy to reduce postharvest losses. The physicochemical properties and the stability of tannins (related to astringency), carotenoids, and their antioxidant activity were assessed during storage. Moreover, in vitro gastrointestinal digestion was conducted to evaluate the recovery index of bioactive compounds, including the oral, stomach, and small intestine stages.

## 2. Materials and Methods

### 2.1. Bars Preparation

Persimmon fruit (*Diospyros kaki* Thunb. cv. ‘Rojo Brillante’) was provided by the Instituto Valenciano de Investigaciones Agrarias (IVIA, Spain) and was not subjected to any deastringency treatment prior to analysis. The persimmons were harvested from a local grove in L’Alcudia, Valencia, Spain, in early December at the ripening stage V [9], which is characterized by an orange color. The persimmons were washed, peeled, and dried in an oven Binder model FD 260 standard (Binder GmbH, Tuttlingen, Germany) at 45 °C for 36 h until they reached a water content of 37 g water/100 g product. Walnuts, hazelnuts, and chia seeds (Hacendado trademark) were procured from a local supermarket.

To prepare the bars, 190 g of dried persimmons, 50 g of walnuts, 50 g of hazelnuts, and 10 g of chia seeds were ground together for 3 min using a Moulinex grinder (Moulinex, Barcelona, Spain) until a homogeneous mixture was formed. Subsequently, bars measuring 10 × 2 × 1.5 cm were prepared. The final formulation of the bars was selected following a series of laboratory tests in which different ingredient ratios were evaluated. The selected bar exhibited the greatest mechanical stability while incorporating the highest possible amount of dried persimmon. The bars were stored at a refrigerated temperature of 4 °C for 4 weeks.

### 2.2. Proximate Composition

Proximate analysis (moisture, ash, fat, protein, sugar, and fiber) was conducted in accordance with the methodology delineated by the AOAC [23]. Carbohydrates were calculated by the difference (Equation (1)), and the gross energy value was obtained by applying Equation (2) [24].
Carbohydrate (%) = 100 − [moisture (%) + protein (%) + fat (%) + ash (%)](1)
Energy (kJ/100 g) = (protein × 16.7) + (fat × 37.7) + (carbohydrate × 16.7) (2)

### 2.3. Mechanical Properties

A TA-XT Plus Texture Analyzer (Stable Micro System, Ltd., Surrey, UK) with a Warner-Bratzler reversible blade (HDP/BS) was used to perform a shear test. The sample was positioned over a slotted blade insert and bisected in the transverse direction. The speed of the test was 1 mm/s and the maximum force (in Newtons) was recorded. Each sample was subjected to six replicates.

### 2.4. Optical Properties

The color was evaluated using a Minolta CM-3600d spectro-colorimeter (Minolta Co., Tokyo, Japan), with the illuminant D65 and the 10° observer serving as the reference to obtain the CIE L*a*b* color coordinates. Each sample was subjected to six replicates. The hue (h*) and chroma (C*) values were calculated using Equations (3) and (4), respectively. The color differences (ΔE*) were calculated using Equation (5), with the initial sample (t 0) designated as the reference point.
(3)$$h*=arctg\frac{b*}{a*}$$
(4)$$C*={\left[({a*}^{2}+{b*}^{2}\right)}^{1/2}]$$
(5)$$∆E*={\left[{\left(∆L*\right)}^{2}+{\left(∆a*\right)}^{2}+{\left(∆b*\right)}^{2}\right]}^{1/2}$$

### 2.5. Soluble Tannin Content (STC)

The STC was determined using the Folin–Ciocalteu colorimetric method, as described by Arnal et al. [25], in an ethanolic extract (extracted with 96% ethanol). The results are expressed in terms of mg of gallic acid equivalent (GAE) per 100 g of sample. Tannin extraction was performed in triplicates.

### 2.6. Total Carotenoid Content (TCC)

TCC was calculated in accordance with the protocol established by Nath et al. [26], using a lipidic extract extracted with acetone and diethyl ether. The results are expressed in mg of β-carotene per 100 g of sample. Carotenoid extraction was performed in triplicate.

### 2.7. Antioxidant Activity (FRAP and DPPH)

Antioxidant activity was measured using the FRAP and DPPH protocols described by Benzie et al. [27] and Matsumura et al. [28], respectively. FRAP results are reported as µmol of Trolox per gram of sample, whereas the DPPH values are expressed as inhibition %.

### 2.8. In Vitro Digestion

The methodology proposed by Brodkorb et al. [29] was used to simulate the biological fate of the ingested substances using an in vitro gastrointestinal tract model. All oral, gastric, and intestinal phases were replicated. The complete set of enzymes required for the analysis was obtained from Sigma-Aldrich (Madrid, Spain). In accordance with the methodology proposed by González et al. [30], the digestive process was conducted in a “Carousel 6 Plus” reaction station (Radleys, Saffron Walden, UK) under controlled conditions at a temperature of 37 °C, with agitation at 150 rpm, in the absence of light, and in an atmosphere of N_2_. The recovery index was calculated to analyze the effect of in vitro digestion on the content of soluble tannins and carotenoids. The recovery index was used to quantify the amount of substance recovered after intestinal digestion. This was achieved by comparing the recovered amount with that of an undigested sample [31]. The recovery index (RI) was calculated using Equation (6):(6)$$Recovery\;index\;\left(\%\right)=\frac{DF}{UDF}\times 100$$
where *DF* (digested fraction) is the amount of bioactive compounds in the digested fraction following the intestinal digestion phase, and *UDF* (undigested fraction) is the amount of bioactive compounds quantified in fresh samples.

### 2.9. Statistical Analysis

Statistical analysis of the data was conducted using the Statgraphics Centurion XVII program, using analysis of variance (ANOVA) and calculating the minimum Fisher’s significant differences at a 95% significance level (*p* < 0.05).

## 3. Results and Discussion

### 3.1. Proximate Composition of the Bars

The proximate compositions of the persimmon-based bars are presented in Table 1. The moisture content was comparable to that reported for fruit bars formulated using date paste [14]. Although Parna et al. [14] introduced small amounts of water to aid the blending process of dates into the paste, our formulation did not require water addition, as the dehydrated persimmon served as a cohesive matrix for the bar.

The bar is a rich source of healthy fats derived from walnuts, hazelnuts, and chia seeds. These unsaturated healthy fats contribute to optimal brain function, are crucial for cardiovascular health, and serve as sustainable energy sources [32]. In addition, the seeds and nuts assisted in maintaining equilibrium between the protein and fiber content of the persimmon-based bar, which exhibited higher levels than other fruit-based bars. Sun-Waterhouse et al. [2] reported protein and fiber contents of 2.74% and 2.54%, respectively, in snack bars containing apple polyphenol extracts. These values are more than twice as low as the protein and fiber values obtained for the persimmon bars in the present study. In date-based bars, the values were 3.70% and 5.51%, respectively, when the cultivar Nabtat Ali was used and 4.06% and 4.47% when bars were formulated with dates from the Sukkari cultivar [14]. All of these values were lower than those of the persimmon bars. In a separate study, Srivastava et al. [33] reported that the guava-orange fruit bars they prepared contained fat in the range of 1.40% to 4.80%, which is significantly lower than the fat content of the bars in this study (22.80%). The persimmon-based bar comprises sugars as the primary carbohydrate source (Table 1). However, because no additional sugars were incorporated during the manufacturing process, the values were lower than those of other fruit-based energy bars. In a separate study, Agahari et al. [33] reported a sugar content of 70.98% in apple bars formulated with added sugar. Parna et al. [14] reported higher carbohydrate values in date-based bars despite the absence of additional sugars in the formulation. In a study by Lucas-González et al. [34], the sugar content of fruit snacks prepared with fig, pomegranate, date, or apricot ranged from 30 to 55%.

For macronutrients, the energy value of the persimmon-based bar was 1640.62 kJ/100 g, which was higher than that reported for other fruit-based energy bars [15]. Therefore, this nutritionally dense and calorically concentrated product can be regarded as a convenient food option for athletes and for individuals with specific dietary requirements, such as individuals following a high-protein diet with no added sugar.

### 3.2. Mechanical Properties

A crucial aspect of developing a novel fruit-based bar is that the product retains its structural integrity and displays no indications of disaggregation. Thus, various hydrocolloids have been considered for the production of fruit-based bars. Dana-Lache et al. [35] evaluated the incorporation of low-acyl and high-acyl gellan into mango-based bars, whereas Ahmad et al. [36] studied the effect of pectin, starch, and ethyl cellulose addition in fruit bars made from papaya and tomato. Dehydrated persimmon proved to be an effective matrix for bar formulation owing to its cohesiveness, obviating the need for additional hydrocolloids.

Table 2 presents the mean values of the maximum force (N) recorded in the shear test conducted on the samples over a four-week storage period. The test enabled the identification of alterations in the mechanical characteristics of persimmon-based bars. The bars exhibited a notable increase (*p* < 0.05) in F(N) values during the initial two weeks of storage. During the third week, these values remained constant. In the final week, it decreased significantly (*p* < 0.05) until it reached a value that was not significantly different from that recorded in the first week (*p* > 0.05). Therefore, the mechanical properties remained consistent after one and four weeks of storage, and no evidence of disaggregation was observed.

The evolution of mechanical properties during storage has been documented in multiple studies. Ibrahim et al. [15] observed a progressive hardening of the date-based bars after 12 days of storage, yielding similar results. Ahmad et al. [36] reported significant differences after 120 days of storage, depending on the hydrocolloid combination. The findings indicated that the incorporation of pectin and starch (at concentrations of 0.5, 1, and 1.5%) resulted in a notable (*p* < 0.05) increase in the hardness of the fruit-based bar prepared from a combination of papaya and tomato. However, the incorporation of starch and ethyl cellulose (at concentrations of 0.5% and 1%, respectively) resulted in a statistically significant reduction (*p* < 0.05).

### 3.3. Optical Properties

As illustrated in Table 2, there was a notable decline (*p* < 0.05) in the L* value during storage compared with week 0. Similarly, the h* value exhibited a significant (*p* < 0.05) reduction with prolonged storage, indicating a shift toward a redder hue. The chromatic purity or chroma, which is related to the value of C*, also demonstrated variation throughout the storage period, exhibiting a significant decrease (*p* < 0.05) with increasing storage time. The ΔE* value reached 15 at the conclusion of the fourth week. The primary color alterations occurred during the initial two weeks of storage; subsequently, most optical parameters exhibited minimal fluctuations (*p >* 0.05). These changes can be attributed to several biochemical reactions, including enzymatic browning due to polyphenol oxidases, non-enzymatic browning such as Maillard reactions, oxidative breakdown of ascorbic acid, and degradation of carotenoids [37].

### 3.4. Soluble Tannin Content (STC) and Antioxidant Activity

The STC and related antioxidant activity of persimmon-based bars stored for four weeks are presented in Table 3.

The STC of the persimmon-based bar was 78.41 ± 1.67 mg GAE/100 g at the outset of the storage period. Given the findings of previous studies examining the correlation between STC and astringency, it can be concluded that the STC values observed in this study fall within the range typically observed for non-astringent persimmon products [8,38]. As González et al. [12] have previously observed, hot-air drying represents an effective technique for reducing STC, enabling the production of non-astringent dehydrated persimmon products derived from astringent fruits that have undergone no preliminary deastringency treatment. The bars formulated in this study serve as a prime example of this phenomenon. During storage, a statistically significant decline (*p* < 0.05) was observed, reaching a final value of 37.25 ± 2.46 mg GAE/100 g after four weeks at 4 °C. As stated by Zhou et al. [39], the storage of dried persimmons results in a reduction of tannin levels, primarily due to the oxidation and polymerization processes, which transform tannins from soluble to insoluble forms. The enzymatic action of polyphenol oxidase (PPO) facilitates the oxidation of soluble tannins, resulting in their polymerization into larger, insoluble molecules. As these molecules precipitate, the astringency and bitterness associated with soluble tannins are reduced. Therefore, the persimmon-based bar did not exhibit astringency issues related to the solubilization of insoluble tannins during storage, in contrast to other ‘Rojo Brillante’ persimmon products, such as spreadables [13]. This represents a promising avenue for the valorization of the astringent ‘Rojo Brillante’ cultivar, which could have significant implications for the wider food industry.

Table 3 presents the findings pertaining to the antioxidant activity of the ethanolic extract as determined by the FRAP and DPPH methods. A notable decline (*p* < 0.05) in antioxidant activity was evident during the initial two weeks, exhibiting a trend similar to that of STC. The observed decline in the antioxidant activity of the persimmon-based bar may be attributed to the insolubilization and oxidation of tannins during storage [7,40].

The persimmon-based bar exhibited higher STC and antioxidant activity than the raw material and dehydrated persimmons. The STC in the dehydrated persimmon was 51.17 ± 11.11 mg GAE/100 g, with a FRAP value of 8.55 ± 0.96 (µmol Trolox/g) and a DPPH value of 69.77 ± 8.65 inhibition %. The increase is due to additional ingredients in the formulation, specifically walnuts and hazelnuts, which have high levels of phenolic compounds. The total phenolic content of walnuts ranged between 1558 and 1625 mg GAE/100 g, whereas hazelnuts have been found to have a lower total phenolic content, with values between 291 and 875 mg GAE/100 g [41]. The phenolic compounds present in chia seeds have been found to range from 53.5 to 71.2 mg GAE/100 g [42].

### 3.5. Total Carotenoid Content (TCC) and Antioxidant Activity

The results indicated that the TCC was higher in the dehydrated persimmon (129.16 ± 0.38 mg β-carotene/100 g) than in the persimmon-based bar (Table 4). However, the formulated bar exhibited higher antioxidant activity than the dehydrated persimmon (1.82 ± 0.39 inhibition %), which can be attributed to the presence of other fat-soluble components in the formulation, including vitamin E in hazelnuts [19], chia [43], and walnuts [44].

The primary cause of carotenoid loss in fruits during storage is oxidation, which is affected by several variables, including temperature, light, and oxygen exposure [45,46]. Song et al. [45] observed a reduction in the TCC of dehydrated pumpkins during storage at 4 °C; they reported that the main reason for significant losses of carotenoids is non-enzymatic oxidative degradation occurring in the presence of molecular oxygen. However, in the persimmon-based bar, TCC remained stable, with no statistically significant differences (*p >* 0.05) detected throughout the storage period. Walnuts and hazelnuts are especially rich in antioxidants, including tocopherols (vitamin E), which contribute to their high antioxidant efficacy [47], these antioxidants help protect carotenoids, such as beta-carotene, from oxidative degradation. Regarding antioxidant activity, the values remained constant, except for the second week, during which a significant increase (*p* < 0.05) was observed.

### 3.6. Effect of In Vitro Digestion on STC, TCC and Antioxidant Activity

Following in vitro digestion, a notable increase (*p* < 0.05) in STC was observed in comparison with the undigested bars, resulting in an RI% value of 180.08% (Table 5). These findings are consistent with those of previous studies on persimmon products [8,30]. The liberation of tannins linked to fiber and proteins in the food matrix and the higher solubility of tannins due to factors such as the pH environment in the gastrointestinal phase may be the reason for the observed increase in STC in the digesta [48]. The antioxidant activity, as determined by the FRAP method, exhibited a similar trend, with values increasing from 11.12 before in vitro digestion to 15.74 µmol Trolox/g after in vitro digestion. Conversely, DPPH assay demonstrated a decline in antioxidant activity, with a reduction in inhibition from 38.32% to 28.87%. Kamiloglu et al. [49] reported a significant negative impact on the antioxidant activity of black carrot, jam, and marmalade after in vitro digestion. Nevertheless, the addition of persimmon flour to muffin formulations resulted in a notable enhancement of antioxidant activity within the soluble fraction of the small intestinal digestate [8].

As previously documented [8], TCC was found to undergo a notable decline (*p* < 0.05) following the in vitro digestive process. This decline was observed in the persimmon-based bar, with the non-digested sample containing 77.87 mg β-carotene/100 g, and the digesta from the small intestine exhibiting a significantly reduced concentration of 7.69 mg β-carotene/100 g. Carotenoids that underwent intestinal micellization before absorption exhibited a decrease in RI below 10% (Table 5). Several factors may influence carotenoid absorption, including (i) food processing, (ii) meal composition, (iii) digestive enzyme activity, and (iv) cross-enterocyte transport efficiency, as described by Desmarchelier et al. [50]. The reduction in TCC content after in vitro digestion may be attributed to the lipophilic properties of the compounds and the matrix effect. The bioavailability of carotenoids in plant-based foods is typically low due to their entrapment within plant cell membranes and dietary fibers. [50]. However, regarding the antioxidant activity evaluated by DPPH in the lipid fraction, a notable increase was observed following in vitro digestion, as evidenced by a statistically significant difference (*p* < 0.05). A comparable trend was observed in muffins fortified with persimmon flour [8].

## 4. Conclusions

The development of persimmon-based energy bars using the astringent ‘Rojo Brillante’ variety represents a novel approach to mitigate substantial postharvest losses of this crop while introducing a nutrient-dense product with strong market potential. Dehydrated persimmon serves as a natural binder, obviating the necessity for added sugars or hydrocolloids, thus maintaining a clean-label appeal. The incorporation of walnuts, hazelnuts, and chia seeds significantly enhanced the nutritional value of the bars, particularly in terms of healthy fats, proteins, and dietary fiber. This makes the bars a balanced option for health-conscious consumers. The bars demonstrate minimal disaggregation and desirable firmness over a four-week storage period. The stability of carotenoids and the controlled reduction of soluble tannins during storage demonstrated the effectiveness of the formulation in controlling astringency and preserving bioactive compounds. The in vitro digestion results demonstrated that soluble tannins are more bioavailable, which lends further support to the health benefits of these compounds. Despite the reduction in carotenoid recovery, the product demonstrated considerable antioxidant capacity following digestion. Therefore, this persimmon-based energy bar represents a sustainable and functional alternative within the snack food industry, addressing both consumer demand for natural and nutritious products and the need to mitigate agricultural waste. Further research should concentrate on the prolongation of the bars’ shelf life, which can be achieved by the assessment of diverse packaging and storage conditions. Furthermore, a sensory analysis should be conducted to identify potential enhancements that could facilitate a successful market launch.

## Figures and Tables

**Table 1 foods-13-03748-t001:** Proximate composition of the persimmon-based bar (*n* = 2).

Macronutrients	(g/100 g)
Moisture	28.83 ± 1.24
Fat	22.80 ± 0.83
Carbohydrate	39.37 ± 1.41
-Sugar	26.60 ± 1.27
-Fiber	5.80 ± 0.38
Protein	7.40 ± 0.63
Ash	1.60 ± 0.09

**Table 2 foods-13-03748-t002:** Maximum force (N), lightness (L*), chroma (C*), hue (h*), and color difference (∆E*) of persimmon-based bars during storage (*n* = 6).

t (weeks)	F(N)	L*	C*	h*	ΔE*
0	7.82 ^a^ ± 0.72	37.24 ^a^ ± 1.26	24.63 ^a^ ± 1.68	75.29 ^a^ ± 0.66	-
1	10.27 ^b^ ± 0.74	32.27 ^b^ ± 0.97	14.84 ^b^ ± 0.66	68.30 ^b^ ± 1.41	11.23
2	12.32 ^c^ ± 1.05	28.46 ^c^ ± 1.09	12.02 ^c^ ± 1.24	64.93 ^c^ ± 1.63	15.79
3	12.98 ^c^ ± 0.70	30.4 ^c^ ± 1.57	12.9 ^c^ ± 0.93	67.07 ^b^ ± 3.19	13.83
4	10.55 ^b^ ± 0.65	29.85 ^c^ ± 1.03	11.68 ^c^ ± 0.52	63.55 ^c^ ± 2.74	15.31

Mean values in a column with different superscript letters differ significantly (*p* < 0.05) according to ANOVA (LSD multiple range test).

**Table 3 foods-13-03748-t003:** Soluble tannin content (STC) and antioxidant activity of persimmon-based bars during storage (*n* = 3).

t (Weeks)	STC (mg GAE/100 g)	FRAP (µmol Trolox /g)	DPPH (Inhibition %)
0	78.41 ^a^ ± 1.67	11.12 ^a^ ± 0.28	90.44 ^a^ ± 1.43
1	70.55 ^b^ ± 3.22	8.46 ^b^ ± 0.55	64.79 ^b^ ± 2.56
2	57.01 ^c^ ± 6.89	6.84 ^c^ ± 0.47	44.85 ^c^ ± 1.33
3	50.61 ^c^ ± 6.05	6.24 ^c^ ± 0.53	43.91 ^c^ ± 4.24
4	37.25 ^d^ ± 2.46	6.54 ^c^ ± 0.45	35.02 ^d^ ± 2.46

Mean values in a column with different superscript letters differ significantly (*p* < 0.05) according to ANOVA (LSD multiple range test).

**Table 4 foods-13-03748-t004:** Total carotenoid content (TCC) and antioxidant activity of the persimmon-based bar during storage (*n* = 3).

t (Weeks)	TCC (mg β-Carotene/100 g)	DPPH (Inhibition %)
0	77.87 ^a^ ± 2.67	32.60 ^a^ ± 5.68
1	75.78 ^a^ ± 11.26	33.61 ^a^ ± 5.54
2	73.72 ^a^ ± 3.34	51.81 ^b^ ± 7.21
3	72.79 ^a^ ± 7.43	40.38 ^a^ ± 3.44
4	69.33 ^a^ ± 4.31	33.61 ^a^ ± 5.54

Mean values in a column with different superscript letters differ significantly (*p* < 0.05) according to ANOVA (LSD multiple range test).

**Table 5 foods-13-03748-t005:** Soluble tannin content (STC), antioxidant activity of the soluble fraction (FRAP-s and DPPH-s), total carotenoid content (TCC), antioxidant activity of the carotenoid fraction (DPPH-c), and RI% of the STC and TCC of the persimmon-based bar after in vitro digestion (*n* = 2).

After In Vitro Digestion	
STC (mg GAE/100 g)	141.21 ± 4.11
FRAP-s (µmol Trolox/g)	15.74 ± 1.03
DPPH-s (inhibition %)	38.32 ± 3.36
RI% STC	180.08
TCC (mg β-carotene/100 g)	7.69 ± 0.74
DPPH-c (inhibition%)	44.23 ± 13.95
RI% TCC	9.87

## Data Availability

The original contributions presented in the study are included in the article, further inquiries can be directed to the corresponding author.

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
