# Peer review of "Valorizing Astringent ‘Rojo Brillante’ Persimmon Through the Development of Persimmon-Based Bars"

_foods, 2024, doi:10.3390/foods13233748_

Round 1
Reviewer 1 Report
Comments and Suggestions for Authors This manuscript is a very well written article. The work is interesting and well organized. Some parts need to be improved before further editing. I have highlighted them below: Comment 1: The researchers found that the TCC remained stable in the lotus-based bars remained stable, with no statistically significant differences (p > 0.05) observed throughout the storage period. Please indicate the storage conditions and the criteria you used to select these conditions. Comment 2: More emphasis should be placed on a comparative evaluation of the results of similar studies to determine that the formulated energy bar contains higher levels of healthy fats, protein and fiber than other fruit energy bars. Comment 3: The way in which the fruit energy bar contributes to the reduction of agricultural waste should be better documented with quantitative data from the production of the astringent fruit variety "Rojo-Brillante". Lines 56-57: The research team found a significant decrease in soluble tannin content in the dried fruit slices after drying at 40 and 60 °C. Further description of the results considering the methods is required. Results and discussion Line 200: Explain in more detail which category of people with special nutritional needs you have mentioned. Line 328: Give recommendations for future research.This manuscript is a very well written article. The work is interesting and well organized. Some parts need to be improved before further editing. I have highlighted them below: Comment 1: The researchers found that the TCC remained stable in the lotus-based bars remained stable, with no statistically significant differences (p > 0.05) observed throughout the storage period. Please indicate the storage conditions and the criteria you used to select these conditions. Comment 2: More emphasis should be placed on a comparative evaluation of the results of similar studies to determine that the formulated energy bar contains higher levels of healthy fats, protein and fiber than other fruit energy bars. Comment 3: The way in which the fruit energy bar contributes to the reduction of agricultural waste should be better documented with quantitative data from the production of the astringent fruit variety "Rojo-Brillante". Lines 56-57: The research team found a significant decrease in soluble tannin content in the dried fruit slices after drying at 40 and 60 °C. Further description of the results considering the methods is required. Results and discussion Line 200: Explain in more detail which category of people with special nutritional needs you have mentioned. Line 328: Give recommendations for future research.
Author Response
This manuscript is a very well written article. The work is interesting and well organized. Some parts need to be improved before further editing. I have highlighted them below:
Comment 1: The researchers found that the TCC remained stable in the lotus-based bars remained stable, with no statistically significant differences (p > 0.05) observed throughout the storage period. Please indicate the storage conditions and the criteria you used to select these conditions.
The persimmon-based bars were stored at a refrigerated temperature of 4 °C for a period of 4 weeks (Line 125 ). The refrigerated storage condition (4 °C) was chosen in order to:
- minimize degradation: refrigeration slows down enzymatic reactions and oxidation processes, which helps in maintaining the stability of bioactive compounds and antioxidant activity in the bars.
- extend shelf life: low temperatures reduce microbial growth and spoilage, helping to preserve the quality of the product over time.
- prevent astringency: as stated in the introduction section (line 67) other authors working with persimmon-derived products, such as spreadables, observed that refrigeration was the optimal method to prevent astringency.
Comment 2: More emphasis should be placed on evaluating the results of similar studies to determine that the formulated energy bar contains higher levels of healthy fats, protein, and fiber than other fruit energy bars.
More emphasis has been made in the comparative evaluation of results (lines 194-195, lines 198-201) regarding:
- healthy fats: a new study (Srivastava et al, 2019) has been added; this study allows to compare the fat content in our study (22.80%) with the guava-orange fruit bars they prepared (1.40-4.80%)
- protein and fiber: the bar contains 7.40% protein and 5.80% fiber, both of which are higher than those found in comparable fruit bars. For example, the protein and fiber levels in snack bars with apple polyphenol extracts were 2.74% and 2.54%, respectively, while in date-based bars, these levels varied but did not exceed 4.06% protein and 5.51% fiber.
- sugar content: although sugars are the primary carbohydrate source in the persimmon bar (26.60 g per 100 g), no extra sugars were added, resulting in a lower sugar content than fruit bars with added sugars. For instance, apple bars with added sugar had a sugar content of 70.98%.
Comment 3: The way in which the fruit energy bar contributes to the reduction of agricultural waste should be better documented with quantitative data from the production of the astringent fruit variety "Rojo-Brillante".
The quantitative data of persimmon production has been added in lines 37-39
Lines56-57: The research team found a significant decrease in soluble tannin content in the dried fruit slices after drying at 40 and 60 °C.
Further description of the results considering the methods is required.
Yes, we found a reduction in tannins after drying, so we used dried persimmon to formulate the bars. When determining the STC (soluble tannin content) in the bars, we found low values for STC because we used persimmon dried at that temperature. We have described the results considering the methods in lines (270-273), and added lines 273-274 for a better understanding.
Results and discussionLine 200: Explain in more detail which category of people with special nutritional needs you have mentioned.
The category of people with special nutritional needs has been added (line 212-213)
Line 328: Give recommendations for future research.
Some recommendations for future research have been included in the conclusion section (lines 376-380)
Reviewer 2 Report
Comments and Suggestions for Authors
I consider that the authors present an interesting work, however, some points will be corrected or justified to get the acceptation for publication
Please see the attached document.

Author Response
Line 10: Please do not use the third plural person (We)
The use of "we" has been avoided (lines 11-13)
Line 26-28: Please add a reference to support the information
A reference has been added (line 29)
Line 36-37: Please add a reference to support the information
A reference has been added (line 41)
Line 45-47: Please add a reference to support the information
A reference has been added (line 51)
Line110: V ripening stage was used based on which scales, use the reference,
A reference has been added (line 115)
Line 115: Please explain what was the reason for using only one formulation; the bar was obtained the first time, or other formulations were developed; please report how it was in visual appearance. What were the parameters for selecting the formulation reported?
The explanation has been added to the manuscript (lines 123-126).The final formulation of the bars was chosen after several laboratory tests in which different ingredient ratios were mixed. The selected bar was the one that was mechanically stable while incorporating the highest possible amount of dried persimmon.
Line 171: The results are well described. However, in the discussion, it would be great if the author reported at least 2 formulations to compare. The author does not report how they chose the formulation reported. I presume that the authors did a sensorial analysis to find the best formulation. It would be of great value if the authors present, if applicable, the evaluations previously carried out to determine the formulation presented, or failing that, the comparison with other formulations or controls is presented.
The explanation has been added to the manuscript (lines 123-126). We did not conduct sensory testing; to choose the formulation, we relied on mechanical properties, selecting the bar that was the most texturally stable.
Reviewer 3 Report
Comments and Suggestions for Authors
This paper presents a study on the development of a novel energy bar based on astringent persimmon fruit (variety “Rojo Brillante”) to reduce postharvest losses. By incorporating walnuts, hazelnuts, and chia seeds, the bar achieves an enhanced nutritional profile, offering healthy fats, proteins, and dietary fiber. The authors analyzed the physicochemical properties, including color, texture, soluble tannins, and carotenoid stability during storage, and evaluated the recovery rate of bioactive compounds through an in vitro digestion model. Overall, this article basically meets the publication requirements of the Foods journal. Minor changes are recommended.
Detailed Issues and Suggestions
Abstract and Introduction
Line 18-19: The abstract includes some unclear statements; for example, “results of the in vitro digestion model” lacks specific details. Consider adding data on changes in bioactive components post-digestion to improve scientific value.
Line 42-52: In the Introduction, while the background on astringency in the “Rojo Brillante” persimmon and the limitations of existing de-astringency treatments is well-addressed, there is insufficient theoretical support for ingredient selection (e.g., why walnuts and hazelnuts were chosen). Consider enhancing this section with scientific rationale for ingredient choice.
Materials and Methods
Line 111-113: The material processing section describes the drying method, temperature, and moisture control adequately, but lacks details on standardization, such as whether drying time was strictly controlled. This could impact reproducibility.
Line 128-129: The descriptions of color, texture, and mechanical property tests lack sufficient details, such as specific parameters set for the texture analyzer. Providing detailed experimental parameters would facilitate replication by other researchers.
Line 152-158: The in vitro digestion model lacks clarity in the procedural steps. It is recommended to list the types, concentrations, and durations of enzymes used in each phase of digestion to clarify the scientific basis of this method.
Results and Discussion
Line 180-181: “The bar is a rich source of healthy fats derived from walnuts (16.67%), hazelnuts (16.67%), and chia seeds (3.33%).” The sources and proportions of healthy fats are given here, but it is not clear based on what these proportions are calculated (e.g., whether it is the proportion of total fat content or the proportion of the entire bar). The expression is not sufficiently clear, and further clarification is suggested.
Line 253-264 & Line 286-292: Data interpretation is somewhat limited. For instance, the authors discuss the reduction in tannin content and carotenoid stability but do not explore possible causes and mechanisms in depth. Adding literature-based discussion of tannin transformation pathways during storage would be beneficial.
L299-312: Regarding the results of bioactive components in vitro digestion, while the authors observe increased tannin bioavailability post-digestion, they lack a mechanistic discussion. Referring to similar findings in other in vitro digestion studies could help provide a reasonable explanation.
Line 178: In table 1, consider including standard deviations to enhance the visual presentation of data. It is also recommended to clearly sample sizes in table annotations.
Line 221-222: It is recommended to clearly indicate sample sizes in table annotations.
Line 250-251: It is recommended to clearly indicate sample sizes in table annotations.
Line 295-296: It is recommended to clearly indicate sample sizes in table annotations.
Line 313-315: It is recommended to clearly indicate sample sizes in table annotations.
Line 322-328: “ Several factors may influence carotenoid absorption, including (i) food processing, (ii) meal composition, (iii) digestive enzyme activity, and (iv) cross-enterocyte transport efficiency, as described by Desmarchelier et al. [47].” The factors affecting carotenoid absorption are mentioned here, but the specific role or influence degree of these factors in this study is not elaborated. Appropriate discussion could be added.
Conclusion
Line 330-346: The conclusion does not fully summarize the study’s innovations and practical implications. Consider highlighting the bar’s contribution to addressing postharvest persimmon losses and discussing its market potential. Suggestions for future research are not specific enough. For example, consider mentioning possible improvements in formulation or processing to optimize the stability and bioavailability of bioactive components.
Language and Formatting
Some sentences are complex, and simplification would improve readability.
There are minor inconsistencies in citation formatting. A final review for formatting consistency in line with journal requirements is recommended.
In summary, this paper has significant potential for publication due to its scientific and practical value. However, addressing the above detailed comments would enhance the paper’s overall quality and rigor.
Author Response
Review Comments
Summary of the Study
This paper presents a study on the development of a novel energy bar based on astringent persimmon fruit (variety “Rojo Brillante”) to reduce postharvest losses. By incorporating walnuts, hazelnuts, and chia seeds, the bar achieves an enhanced nutritional profile, offering healthy fats, proteins, and dietary fiber. The authors analyzed the physicochemical properties, including color, texture, soluble tannins, and carotenoid stability during storage, and evaluated the recovery rate of bioactive compounds through an in vitro digestion model. Overall, this article basically meets the publication requirements of the Foods journal. It is recommended to be accepted after minor revisions.
Detailed Issues and Suggestions
Abstract and Introduction
Line 18-19: The abstract includes some unclear statements; for example, “results of the in vitro digestion model” lacks specific details. Consider adding data on changes in bioactive components post-digestion to improve scientific value.
Numerical data have been added following the reviewer’s suggestion (line 20) .
Line 42-52: In the Introduction, while the background on astringency in the “Rojo Brillante” persimmon and the limitations of existing de-astringency treatments is well-addressed, there is insufficient theoretical support for ingredient selection (e.g., why walnuts and hazelnuts were chosen). Consider enhancing this section with scientific rationale for ingredient choice.
The scientific explanation of why those specific ingredients are chosen is explained in lines 93 to 99.
Materials and Methods
Line 111-113: The material processing section describes the drying method, temperature, and moisture control adequately, but lacks details on standardization, such as whether drying time was strictly controlled. This could impact reproducibility.
Drying time has been added in line 117
Line 128-129: The descriptions of color, texture, and mechanical property tests lack sufficient details, such as specific parameters set for the texture analyzer. Providing detailed experimental parameters would facilitate replication by other researchers.
More information about mechanical and optical properties methodology has been added ( lines, 135 and 142).
Line 152-158: The in vitro digestion model lacks clarity in the procedural steps. It is recommended to list the types, concentrations, and durations of enzymes used in each phase of digestion to clarify the scientific basis of this method.
The in vitro digestion model followed in this research is the harmonized protocol suggested by the INFOGEST network, as outlined in Brodkorb et al. The types, concentrations, and durations of the enzymes in each phase of digestion are those specified in the article mentioned above. The only changes introduced to this protocol are the use of darkness and an oxygen-free atmosphere to prevent oxidation of the bioactive compounds, which are precisely the focus of the in vitro study.
Results and Discussion
Line 180-181: “The bar is a rich source of healthy fats derived from walnuts (16.67%), hazelnuts (16.67%), and chia seeds (3.33%).” The sources and proportions of healthy fats are given here, but it is not clear based on what these proportions are calculated (e.g., whether it is the proportion of total fat content or the proportion of the entire bar). The expression is not sufficiently clear, and further clarification is suggested.
The numbers refer to the percentage of each ingredient in the bar. However, as pointed out by the reviewer it can lead to misunderstanding. Therefore, the numbers have been deleted since the quantity of each ingredient is already mentioned in the M&M section.
Line 253-264 & Line 286-292: Data interpretation is somewhat limited. For instance, the authors discuss the reduction in tannin content and carotenoid stability but do not explore possible causes and mechanisms in depth. Adding literature-based discussion of tannin transformation pathways during storage would be beneficial.
Literature-based discussion about STC and carotenoid changes during storage has been added in lines 276-284 and 314-317.
L299-312: Regarding the results of bioactive components in vitro digestion, while the authors observe increased tannin bioavailability post-digestion, they lack a mechanistic discussion. Referring to similar findings in other in vitro digestion studies could help provide a reasonable explanation.
The mechanism responsible for increased tannin bioavailability post-digestion is provided in lines 329-332. It is mainly related to the solubilization during the digestion of polyphenols linked to protein and fiber in the native structure.
Line 178: In table 1, consider including standard deviations to enhance the visual presentation of data. It is also recommended to clearly sample sizes in table annotations.
Standard deviations have been added in Table 1.
The sample size (n=X) has been added in the table annotations
Line 221-222: It is recommended to clearly indicate sample sizes in table annotations. The sample size (n=X) has been added in the table annotations
Line 250-251: It is recommended to clearly indicate sample sizes in table annotations.
The sample size (n=X) has been added in the table annotations
Line 295-296: It is recommended to clearly indicate sample sizes in table annotations.
The sample size (n=X) has been added in the table annotations
Line 313-315: It is recommended to clearly indicate sample sizes in table annotations.
The sample size (n=X) has been added in the table annotations
Line 322-328: “ Several factors may influence carotenoid absorption, including (i) food processing, (ii) meal composition, (iii) digestive enzyme activity, and (iv) cross-enterocyte transport efficiency, as described by Desmarchelier et al. [47].” The factors affecting carotenoid absorption are mentioned here, but the specific role or influence degree of these factors in this study is not elaborated. Appropriate discussion could be added.
Lines 353 to 356 have been added to explain the factors affecting carotenoids absorption.
Conclusion
Line 330-346: The conclusion does not fully summarize the study’s innovations and practical implications. Consider highlighting the bar’s contribution to addressing postharvest persimmon losses and discussing its market potential. Suggestions for future research are not specific enough. For example, consider mentioning possible improvements in formulation or processing to optimize the stability and bioavailability of bioactive components.
The conclusion section has been modified ( lines 362-363 and 377-381) following the reviewer´s suggestions.
Language and Formatting
Some sentences are complex, and simplification would improve readability.
The work has been reviewed by a native English speaker.
There are minor inconsistencies in citation formatting. A final review for formatting consistency in line with journal requirements is recommended.
The references have been carefully reviewed
In summary, this paper has significant potential for publication due to its scientific and practical value. However, addressing the above detailed comments would enhance the paper’s overall quality and rigor.
Round 2
Reviewer 2 Report
Comments and Suggestions for Authors
Thak you for your responses